# A Secure Communication System for Constrained IoT Devices—Experiences and Recommendations

**DOI:** 10.3390/s21206906

**Published:** 2021-10-18

**Authors:** Michał Goworko, Jacek Wytrębowicz

**Affiliations:** Institute of Computer Science, Faculty of Electronics and Information Technology, Warsaw University of Technology, ul. Nowowiejska 15/19, 00-665 Warsaw, Poland; michal.goworko.stud@pw.edu.pl

**Keywords:** Internet of Things, IoT, information security, elliptic key cryptography, IoT certificate, cryptographic certificate

## Abstract

The Internet of Things networks connect a large number of devices and can be used for various purposes. IoT systems collect and process vast amounts of often sensitive data. Information security should be the key feature of an IoT network. In this paper, we present the IoT-Crypto—secure communication system for the Internet of Things. It addresses IoT features, such as constrained abilities of devices, needs to reduce the volume of the transmitted data and be compatible with the Internet. IoT-Crypto introduces an innovative, lightweight certificate format and trust model based on real-world business relations. It also specifies secure communication protocol, which uses underlying encrypted DTLS connection. This paper presents IoT-Crypto in the context of comparable solutions, discusses its distinctive features and implementation details. Results of tests and experiments performed in the IoT-Crypto network confirm that it works correctly and securely. Test network was also used to ascertain the suitability of encoding standards and BLE IPSP profile for the IoT. Directions of future work were discussed based on those results.

## 1. Introduction

The Internet of Things (IoT) is the concept that entails connecting multiple devices within a single network. They cooperate and together carry out assigned tasks. The IoT concept has been swiftly developing and gaining popularity in recent years. It is mainly due to the progressing miniaturisation of electronic devices, declining prices, and increasingly accessible Internet access. It is projected that by 2025 there will be more than 30 billion IoT devices in use, compared to the estimated 13.8 billion in 2021. Most of them are, by design, directly or indirectly (through some proxy or gateway) connected to the global Internet network.

IoT encompasses numerous and varied classes of applications. The most popular are smart home systems, industrial sensor networks, security systems, weather monitoring solutions, and health monitoring systems. The last listed application underlines that information security is the most crucial characteristic required in every IoT solution. IoT systems collect and process vast amounts of data. It should be stored and transmitted securely and be made available only to the entitled individuals, institutions, and systems.

The performance of IoT devices is often purposefully limited to consume less power (which reduces operational cost and makes it possible to use battery supply) and lower manufacturing costs. As a result, IoT devices can often be described as constrained. It is because their lower capabilities result from deliberate choice rather than the currently available state of technology.

Another aspect of the IoT networks is their diversity. It stems from various possible applications and many communication protocols, wireless communication standards, CPU architectures, and programming languages used. IoT network consists of several types of devices—constrained IoT devices grouped into segments served by IoT gateways and high-performance servers collecting and processing the data. Chosen solutions must work correctly and efficiently across the entire spectrum of utilised devices and software.

This paper aims to present key aspects of the design and experiences from implementing and operating IoT communication and security solution named IoT-Crypto. The solution is a continuation of our previous work described in [1,2]. The main contributions of this work are:custom-built IoT solution, designed with an emphasis on security,innovative trust model based on business relations,lightweight IoT certificate format,performance analysis of the overhead of the designed security solution.

The rest of the paper is structured as follows: we discuss related work in Section 2; Section 3 presents the proposed solution, Section 4 describes implementation details and software operation on particular types of devices, Section 5 presents the test network, selection of obtained measurements, and the performance analysis. Finally, Section 6 concludes this work and sets the direction of future work.

## 2. Related Work

The basis for secure communication in IoT systems is enrolling cryptographic material into end devices and binding their identities with those materials. Thanks to the recent advances in elliptic curves cryptography and the growing complexity of electronic circuits used in constrained IoT devices, asymmetric cryptography became an appealing solution for securing IoT systems. A digital certificate is needed to bind trustfully an identity with a public key. Widely used key distribution mechanisms, i.e., Kerberos, Radius, Public Key Infrastructure (PKI), and certificate formats (i.e., X.509, OpenPGP), serve well communication between powerful devices with a human behind. The research of suitable solutions for IoT constrained devices is ongoing.

Raza et al. [3] have attempted to build a key management architecture for the resource-constrained IoT. They propose a key server, called trust anchor, to distribute pre-shared symmetric or raw asymmetric keys between communicating parties. The proposed interactions with the trust anchor minimise message exchange, comparing to Kerberos, Radius or PKI mechanisms. They argue that PKI is not always suitable for constrained environments due to the size of standardised certificates and the lack of economical methods of developing PKI with globally trusted certificates. They built an experimentation environment based on the CoAP, DTLS, UDP, IPv6, 6LoWPAN, and IEEE 802.15.4 protocols for performance measurements. They analysed energy consumption, memory and time overhead related to symmetric key management and cryptography operations. Moreover, they proved that their implementation is resistant to some Denial of Service (DoS) attacks.

In 2016, IETF published RFC 7925 that define TLS/DTLS IoT profiles for IoT. The main goal of creating that document was to protect CoAP messages using DTLS. Eventually, it defines the way of applying TLS or DTLS on constrained IoT devices. IETF still leads some work on solutions for asymmetric key usage by such devices. As the new DTLS version 1.3 is proposed [4], the IoT profile of DTLS is also updated [5]. That IETF draft also specifies that certificates must be of type X.509 v3 and gives some requirements for its fields. Moreover, the draft gives some recommendations on optimising certificate revocation checks and the size of exchanged certificates. Another IETF draft [6] defines a method that uses CoAP as a transport for the Certificate Management Protocol in its lightweight version [7].

One of the first work on a lightweight implementation of an X.509 certificate is presented by Forsby et al. [8]. They define a minimum set of certificate fields and define fixed values for some of them, which allows to omit them when the certificate is used inside a sub-network with constrained IoT devices. Next, they compress the certificate using the CBOR format (RFC 7049). This solution enables considerably reduce the size of certificates, up to ~93%. The downside is the need to transform X.509 certificates on the edge of the IoT network. The unquestionable advantages are the small size of certificates, the reduced power consumption of IoT devices and compatibility with the X.509 standard.

One of the first architectures of certificate management infrastructure dedicated to IoT is proposed in [1]. The paper gives the premises and reasons for a dedicated infrastructure based on asymmetric cryptography. The doubt—is the certificate issuer credible to us—is the main reason to build a certificate management infrastructure dedicated to IoT systems. An answer for such a doubt is that the certificate should be signed by a party we trust in business relations. Thus, it should be possible to have several signatures that confirm the identity of IoT devices and services, for example, issued by manufacturer or designer, system integrator, system maintainer, and system owner. Those parties should support their Identity Certificate Servers (ICS) that enable information exchange with infield ICSes of the IoT system.

The survey of key bootstrapping protocols based on asymmetric keys in the IoT systems by Malik et al. [9], gives the state of the art and points challenges of implementing such methods. The term bootstrapping is used to refer to the generation and exchange of keying materials between not associated devices. The authors have classified the bootstrapping protocols based on the key delivery method, the underlying cryptographic primitive, and the authentication mechanism. They expect that the less explored schemes, as self-certified and certificateless schemes, need to be implemented and evaluated in IoT systems. They point, as further research directions, exploring the optimisation of asymmetric cryptographic protocols, the usage of blockchain, hardware-based and post-quantum cryptography to support key bootstrapping.

The process of certifying a public key is called enrolment. An enrolment protocol for constrained IoT devices was proposed by He et al. [10]. The protocol, called Indraj, works on top of CoAP, DTLS, and UDP. It can be considered a lightweight version of the Enrollment over Secure Transport (EST) protocol (RFC 7030), which works over HTTP, TLS, and TCP. Both EST and Indraj support client-generated public/private key pairs as well as key pairs generated by a public Certificate Authority (CA). Indraj uses the lightweight implementation of an X.509 certificate defined in previous work [8] by the same team. Next, the team improved the Indraj protocol, see Höglund et al. [11]. They call the newer version of the protocol PKI4IoT and the certificate structure XIOT. They propose to standardise XIOT as the third version of X.509 to enable XIOT processing by the widespread PKI servers. However, it is a bit confusing as the X.509 version 3 certificate format is already standardised by ISO/IEC, ITU-T, ANSI X9, and its internet profile is defined in RFC 5280. Their solution handles multiple root CA certificates pre-installed. However, they admit that further investigations on more efficient solutions for CA discovery and querying are needed.

The idea to enable the transport of EST payload via CoAP also motivates the ongoing work on an IETF draft [12]. The CoAP Block-Wise Transfer mechanism is proposed to fragment long EST messages, and only certificate-based client authentication is considered.

Bradbury et al. [13] used a certificate structure, similar to XIOT, to build a trust system architecture for constrained IoT devices. The trust system supports the selection of edge nodes for offloading computation tasks from constrained devices. The difference between XIOT and Bradbury’s certificate is that there is a stereotype tags field that carries trust data in place of optional extension fields.

A recent attempt to define a public-key management infrastructure for IoT is described by Belattaf et al. [14]. Their approach allows several architectures for certification authentication: hierarchical, Web-of-trust based, threshold, and others. The infrastructure is based on cooperating communities, while each community manages certificates following its policy and delegates an object to exchange authentication requests. The object can be a certification authority server or a Web-of-trust key server. The authors defined a specific certificate structure optimised for constrained IoT devices and an algorithm for the object selection. They executed a set of simulations to analyse the performance parameters of their infrastructure (storage and communication cost in the function of network size and the number of malicious nodes). They proved the better performance of the proposed solution than mentioned before the PKI4IoT protocol with the XIOT certificates. However, the description of their infrastructure is a bit unprecise.

## 3. Proposed Solution and Its Distinctive Features

### 3.1. Overview and General Architecture

IoT-Crypto is an example of an IoT system. It contains several custom-designed mechanisms and features. They were developed to tackle several already mentioned challenges connected with the operation of the IoT networks.

The general architecture of the IoT-Crypto system (Figure 1) is based on already existing solutions, both those described in the scientific literature and those commercially available. Many of them implement the three-layered architecture. The architecture has proven to be versatile enough to be suitable for a wide range of applications. It also makes the network structure and the division of responsibility between layers clear and readable. IoT-Crypto network, therefore, consists of cloud, gateway, and IoT devices layers.

The functions usually associated with the Internet of Things are performed by several IoT devices grouped into subnets. IoT devices can perform one or many of the three tasks: executing actions, collecting data (often sensor readings), and generating alerts. The last two functions are very similar. The difference between them is that collecting data may be implemented as a synchronous operation, while generating alerts based on those same data is always asynchronous.

IoT devices grouped into a single subnet can communicate with the servers in the cloud layer through a gateway device. The role of gateways in the network is crucial. In general, they act as communication proxies. More specifically, this role involves translating protocols and supporting communication using varied standards. Gateway devices also perform most of the workload associated with the encryption and checking cryptographic relations of trust between parts of the network. It means that the IoT devices are responsible for performing tasks essential from the business point of view and only for a minimal amount of technical—network and cryptographic—workload.

The cloud layer is responsible for overseeing the entire network, collecting, processing, and storing the data. It consists of one or more server applications. Cloud layer—API and GUI exposed by those applications—is the only element visible to the outside users and external systems.

It is assumed that a single company manages the IoT-Crypto network. It may have business relations with other companies managing their networks. IoT-Crypto solution enables those companies to cooperate. The entire network or part of it may be shared between them. Cooperation may occur on various layers of the network. IoT devices may connect to the gateways and gateways to the cloud servers belonging to the business partner. According to the business agreements, cloud servers may exchange data, sensor readings, and information about network structure. The scope of cooperation can be easily modified, or it may be cancelled altogether. All those mechanisms are based solely on cryptographic operations and identities—the cryptographic relation of trust models business agreements.

### 3.2. Distinctive Security Features

Unlike some of the already mentioned examples of IoT solutions, the IoT-Crypto system is completely security-oriented and employs security by design principle. It would be impossible to deploy it with encryption and security features disabled because they are intertwined with the network structure and mechanisms. Each of the devices possesses a unique cryptographic identity. What distinguishes it from comparable solutions is the design of the identity. It is composed of a cryptographic certificate of a custom, IoT-specific design and dedicated trust relations structure. Cryptographic identities are used for authentication, authorisation, and encryption, which is uncommon among comparable solutions. In many of them, only the authentication mechanism uses the identity. In those solutions, authorisation is often separated from the cryptographic mechanisms. While encryption is still employed, it may use a web server certificate with no relation to the identities of communicating parties. The design of custom cryptographic identity and the model of the cryptographic trust relationship are therefore determining and innovative characteristics of the IoT-Crypto solution.

As already mentioned, custom cryptographic certificate format is the basis of the cryptographic identity of each device in the IoT-Crypto network. Design decisions concerning the certificate affect many further aspects of the network operation and communication between devices. The use of a custom format designed specifically for the IoT allowed great freedom to shape and optimise IoT network functioning. Existing and commonly used formats (such as X.509) can, of course, be used in IoT networks. They are, however, not well suited for this purpose. Nevertheless, various features and ideas of those formats were used to create the IoT-Crypto certificate format.

IoT-Crypto certificates have a small size. Depending on the encoding, a single certificate may be as small as 232 bytes without compression. This quality makes storing and processing on constrained devices less demanding and more energy-efficient. It also reduces the amount of network traffic. The certificate contains two cryptographic elliptic keys. Keys are relatively short, as elliptic keys can be only two times longer than symmetric keys offering the same level of security. One of them serves for session key generation and the second one for digital signatures. Approach with two separated keypairs has numerous justifications. It reduces the risk of a successful attack. Separating keys is also required by law or recommended in some jurisdictions for specific applications. The certificate also contains a unique id, the subject’s name, creation and expiration timestamps and version number. Each certificate may possess signatures generated by other parties. Third-party signatures are associated with the certificate but are not strictly part of it. Therefore, each certificate is self-signed to ensure its integrity and prove possession of the corresponding private key. Additional signatures may be stored alongside the certificate on the device they pertain to or may be known only to the verifying party and stored separately. The only condition is that they must be available during certificate verification. This mechanism shows similarities with the OpenPGP PKI.

There are two most common formats of a cryptographic certificate. X.509 is by far the most popular and most widespread of them. It is also widely used in IoT networks, for example, in solutions offered by some of the biggest IT companies, such as Google, Microsoft, and Amazon. Its main downside from an IoT point of view is its relatively huge size and complexity. X.509 was designed for different purposes than IoT solutions. For example, to assert the identity of web servers on the Internet. In that case, the certificate is valid only when used together with the specific domain name. IoT devices are a very different case. As a result, when the X.509 format is used in IoT networks, some certificate fields must be ignored, or their meaning is changed. Documentation of Google Cloud IoT Core goes as far as to mention that it does not verify the certificate’s subject. X.509 certificates can be used in IoT networks but are not well suited to act as the cryptographic identity of the IoT devices.

X.509 certificates are issued and signed by a trusted third party called Certification Authority. Trust relation is strictly hierarchical. The certificate is verified as trusted when one of the certificated higher in the hierarchy is marked as trusted. This model is very rigid and does not correspond with the business side of real-world IoT applications. When companies cooperate, the relationship of trust is mutual. Implementing trust relations using the third-party model for those pre-existing business relations is problematic. X.509 certificates can be used in IoT networks but are not well suited for them and may cause technical difficulties. A more flexible trust model, and a certificate format tailored for the IoT needs, should therefore be used.

OpenPGP is the other popular format of the cryptographic certificate. The Web of Trust model represents trust relations in that solution. This term describes distributed and decentralised trust model, which constitutes an alternative to the fully centralised X.509. There is no distinguished third party that issues certificates. Instead, any user may generate a self-signed certificate. Any other party may then express its trust by signing this certificate. Trust relation is transitive. The certificate is deemed trusted by the verifying party if there exists some chain of certificates linking them.

The trust model of the IoT-Crypto system is inspired by both X.509 and OpenPGP. Relation of trust is not transitive, as in X.509 model. It is decentralised to correspond with business relations between companies—owners and operators of IoT networks. Many other parties may then sign each certificate. A certificate is trusted when at least one of the signatures is issued by a trusted party. IoT-Crypto trust model allows dynamic changes of the scope of cooperation between companies by modifying the list of trusted certificates and the list of signatures associated with certificates.

The IoT-Crypto certificate format and trust model are key features of the entire solution. They are custom-designed for the IoT networks, constrained devices and the business side of network operation, which helps construct a secure IoT network structure, communication protocols and autoconfiguration mechanisms built upon them.

### 3.3. Autoconfiguration Mechanisms Related to the Security Features

IoT networks consist of numerous devices and continuously collect and process data. The size of the network and a large number of devices make manual configuration and maintenance error-prone or even infeasible. IoT networks should therefore contain autoconfiguration mechanisms and be able to operate autonomously. IoT-Crypto solution contains said autoconfiguration and autonomous operation mechanisms. That includes:detecting and initiating new devices in the network,closing connection with devices that are no longer trusted,periodically fetching data from sensors,periodically changing encryption keys,checking current and storing historical network structure.

In a network without those mechanisms administrator would be responsible for checking the credibility of the devices. All of those mechanisms require certain cryptographic features to be present that allow secure operation and replace manual bootstrapping and configuring the network. It is important to note that IoT gateway performs a key role in all those mechanisms. In its capacity as an intermediary, it can:verify trust relations between devices,monitor connection with IoT devices,translate protocols and relay information between wireless and cable connections,perform load-intensive operations, relieving IoT devices of some responsibilities.

## 4. Implementation Details

### 4.1. Cryptographic Details and Certificate Format

IoT-Crypto software was written mainly in the C programming language. There are several reasons behind this choice. C code can be compiled and run on virtually all hardware platforms, including microcontrollers, constrained ARM and x86 devices and powerful servers. It would be a severe handicap to maintain parts of code responsible for cryptographic operations written in several programming languages and using language-specific libraries. Mbed TLS open-source C library is used as a base for the software. It is actively developed and maintained. Its main distinctive feature is having very few external dependencies. It is possible and easy to replace them if not available on a specific platform. Therefore, software based on the Mbed TLS library will run on any device, thus fulfilling the requirements mentioned in the introduction.

IoT-Crypto certificate is implemented as C structures, as shown in Figure 2. It is an example of a minimal data structure that wraps public keys in the IoT network. It is much lighter than X.509 and OpenPGP certificates and still contains all the necessary information. It may be extended in the future, thanks to version numbering. As mentioned before, it wraps two separate keys used for encryption and signing.

The first of the keys (“ecdhKey” field) is used during cryptographic key exchange. It is based on the elliptic curve Curve25519. It is 256 bits long and provides security comparable to the 128-bit symmetric cypher. The symmetric session key is computed using Elliptic-curve Diffie–Hellman key agreement protocol. The second key is (“ecdsaKey”) to generate self-signature, sign other certificates, and any other binary or text data. This key is 512-bit long and is based on the elliptic curve secp256r1. This curve is recommended (according to RFC5480) when 128-bit security is required. It was impossible to use the same type of key as for key exchange. Elliptic curve digital signature algorithm (ECDSA) implementations for Curve25519 are very rare. It was optimised for ECDH. It is possible to use it for ECDSA, but only a few libraries allow that. Mbed TLS is not one of them. The certificate verification consists of the following steps:Expiration date verification.Self-signature validity verification.Searching for a signature assigned to the certificate and issued by a trusted party.

The certificate is only treated as trusted if all the steps are successful. Even then, the connection may be terminated if the certificate is revoked.

A solution comparable to the IoT-Crypto certificate was presented in the already mentioned work by Höglund et al. [11]. Assumptions made about the structure and requirements of IoT networks are similar to those described in Section 3. The design of the XIOT certificate is, therefore, akin to the IoT-Crypto certificate. There are, however, several key differences resulting from differing decisions made in the early design stages.

XIOT certificate is compatible with X.509 format, while IoT-Crypto certificate is not. The main advantage of preserving compatibility with the most popular certificate format is the possibility of using already existing software, hardware, and algorithms with little or no modifications. However, that approach causes some significant problems. Certificates must be converted to the standard X.509 when used by incompatible devices. Trust relations cannot correspond with IoT network design and business relation structure because it must follow centralised X.509 PKI requirements. XIOT certificate must also support X.509 features not used in the IoT networks, such as some extension types.

XIOT certificate contains only one public elliptic key without separating the signing and encryption keys. It is sufficient and reduces the size in comparison with the IoT-Crypto certificate. The advantages of using two separate keys were already described, but it is not strictly necessary for building a secure IoT-Crypto network.

The last of the analysed differences between the two formats concerns the communication protocol used for certificate exchange and communication. The main difference is that XIOT uses already existing and proven protocols, such as CoAP (lightweight HTTP counterpart), while the IoT-Crypto solution utilises custom-designed UDP-based protocols.

### 4.2. Protocol Stack

IoT-Crypto uses the IP protocol in the network layer of the TCP/IP protocol stack. It may work with both IPv4 and IPv6. They may be mixed within the same IoT network as long as network visibility using the same protocol version is ensured between directly connected devices. Any protocols and standards may be used in the network access layer.

MbedTLS library [15] operates on the border between transport and application layers of the TCP/IP stack. As already mentioned, IoT networks may contain constrained devices. UDP connectionless protocol is much better suited for them than TCP. It introduces less overhead (memory and processor usage, number of transmitted bits), lower latency and generally puts less strain on the device. MbedTLS implementation of DTLS is therefore used as a UDP alternative for the TCP-based TLS. DTLS introduces only certain required features of the TCP protocol (retransmission of lost packets and error correction). The IoT-Crypto library allows writing software running in the application layer of the TCP/IP stack.

### 4.3. IoT-Crypto Device Software

Software of the IoT device in the IoT-Crypto system was designed to be very simple (see software operation diagram in Figure 3). After the device power on, it awaits an initialisation message broadcasted by the gateway. This message contains information about the gateway—id and network address. IoT device may then connect to the gateway and perform key negotiation. After successful key negotiation, the device enters event handling mode and processes synchronous (request/response) messages received from the gateway, i.e., sensor reading requests and commands. The IoT device may also send asynchronous (spontaneous) messages to the gateway, e.g., alerts based on the sensor readings. These two tasks are running simultaneously in separate threads. IoT-Crypto library provides the framework for writing the device’s software. The developer is responsible only for providing either one or both message handler methods and the asynchronous message sender method.

### 4.4. IoT-Crypto Cloud Server Software

IoT-Crypto cloud server is located on the opposite side of the network from IoT devices. Its software performs two main tasks:it is a typical business application—exposing HTTP API and processing requests received from users and external systems, thus enabling interactions with IoT-Crypto network,it is IoT-Crypto cryptographic application that securely communicates with other devices in the network, validates certificates and signatures.

As already mentioned, the core of IoT-Crypto software (especially the parts responsible for the cryptographic operations) is written in C and based on the MbedTLS library. Implementing core cryptographic functionality in other languages would create compatibility issues and the problem of maintaining two versions simultaneously. On the other hand, the C language is ill-equipped to create business applications exposing HTTP API and using the database. IoT-Crypto cloud server software is divided into two parts (as shown in Figure 4) in order to solve this problem.

The first of them is a business application written in Scala and connected with the Postgres database. Language and database choices were arbitrary. Any high-level programming language with good support for creating APIs and any relational database system could have been chosen. The second part of the server software is an application written in C and based on the MbedTLS library, the same as IoT device and gateway software.

Two parts of the server software operate as separate applications and communicate using internal HTTP API. The business application exposes HTTPS REST API, allowing users and external systems to communicate with and access resources in the IoT network. The cryptographic application exposes API for IoT gateways. It uses custom IoT-Crypto communication protocols that will be described later.

### 4.5. IoT-Crypto Gateway Software

The IoT gateway is a crucial element of the IoT-Crypto network. It can handle connections with multiple IoT devices. Each gateway is connected to a single cloud server and is an intermediary between constrained IoT devices and the server. IoT devices connected to the single gateway form a subnet. Various communication protocols and wireless standards may be used simultaneously in the IoT subnet. The gateway must handle all of these and perform protocol translation when needed. IoT-Crypto network gateway is often a point of contact of IPv4 and IPv6 networks using Wi-Fi, BLE and Ethernet standards in the network access layer.

Gateway software is multithreaded and is continuously performing numerous tasks (see software operation diagram in Figure 5). After connection with the cloud server is successfully made, the gateway starts the message handling loop. It receives and handles messages received from the server, including data requests, commands to be forwarded to the devices and certificate revocation messages. Then the gateway begins broadcasting advertisement messages for IoT devices. That way, the network address and port do not have to be manually configured, and device detection is automatic. The second task is a key negotiation service for IoT devices. Verifying the device and exchanging keys is necessary before transmitting any application data. A separate message handling loop is then started for every connected IoT device.

The gateway is located in the middle of the IoT network. That property makes it perform a vital role in all autoconfiguration mechanisms described in Section 3.3. Gateways perform nearly all tasks needed for the network operation, leaving only the most basic tasks of collecting data and performing actions to the IoT devices. It allows the source code of the IoT devices software to be short, have a simple structure, and the resulting software to have low hardware requirements.

### 4.6. IoT-Crypto Communication Protocol

Communication in the IoT-Crypto network is based on the TCP/IP protocol stack with the UDP in the transport layer and MbedTLS implementation of DTLS protocol on the border between transport and application layers. The application layer consists of two custom protocols: IoT-Connection-Protocol and IoT-Transmission-Protocol. Comparable solutions, such as XIOT, often use CoAP or HTTP in that layer. Custom protocols have the advantage of being tailor-made and optimal for the specific application. It allows to avoid unnecessary transmission overhead and optimise the size of the messages. Already existing all-purpose protocols have the advantage of being well tested and documented. Implementations of those protocols are also readily available.

The underlying UDP connection is secured using DTLS and uses cypher suite TLS_PSK_WITH_CHACHA20_POLY1305_SHA256 (defined in the RFC 8439 [16]). It is a suite of cryptographic algorithms consisting of SHA256 hash function, Poly1305 message authentication code and ChaCha20 symmetric stream cypher. According to the specification, it can be up to three times faster than cypher suites that use AES cypher. Cryptographic key exchange is performed on the IoT-Crypto level, and DTLS uses the obtained key as a pre-shared key (PSK).

IoT-Connection-Protocol is responsible for establishing a secure connection between devices. This objective is achieved by employing elliptic key exchange, verification and calculating a common session key. Cloud server exposes a key negotiation service for gateways, and the gateway exposes analogic service for IoT devices. The session key derived from the elliptic Diffie–Hellman key exchange would be the same for each connection between the same devices. Forward security is achieved by making one side of the transmission use ephemeral keys. Less constrained of the two connecting devices is responsible for generating the ephemeral key pair (gateway when connecting with IoT devices and cloud server when connecting with gateways). Protocol messages are presented in Figure 6.

The gateway is actively advertising its presence by periodically sending a broadcast message containing its id, key negotiation service port and transmission service port. IoT device detects the initial message and connects with the appropriate port. The communication is protected with a pre-shared key, common in the entire network. It is not strictly necessary, as the key exchange can be safely performed over a public channel. PSK-based encryption helps to avoid plain-text communication and provides a basic level of security against eavesdropping attacks. Only 1639 bytes need to be transmitted during key exchange. Of those, 608 bytes correspond to the two certificates of connecting devices. A minimal amount of exchanged data and low protocol overhead are the crucial advantages of custom formats and protocols of the IoT-Crypto solution.

Once the symmetric key is established, the IoT device uses it to connect with the gateway’s transmission service as a PSK. The gateway exposes it on a separate port. This service uses IoT-Transmission-Protocol, the second of the custom IoT-Crypto protocols. It is designed to handle three identified applications of IoT devices: collecting data, executing actions and generating alerts. Protocol messages are presented in Figure 7. There are five types of them:data request and data response messages—for handling sensor readings,command execution request and command execution status messagesalert message—for handling alerts generated by IoT devices, e.g., based on sensor readings,error message—for handling malfunctions and unexpected situations.

IoT-Connection-Protocol and IoT-Transmission-Protocol are located in the application layer of the TCP/IP protocol stack. They were custom-designed for the IoT network operation. It made them have a small protocol overhead and be less complicated than all-purpose solutions. Protocol messages are serialised to binary format according to CBOR standard and transmitted over DTLS connection with underlying UDP datagram communication. CBOR encoding test results and analysis of protocol and encryption overhead are presented in the following chapter.

There is a plethora of messaging protocols proposed for IoT systems, e.g., MQTT, CoAP, STOMP, XMPP, WAMP, AMQP, DDS, OPC UA. Only a few of them can work over UDP, i.e., CoAP, MQTT-SN, DDS. Any of them can offer some appealing features for a system designer. However, any of them implies some resources consumption (memory, processor, transmission bandwidth). Thus, the more efficient implementations for constrained devices are custom-designed and based on UDP. IoT-Crypto is a good template for a small or medium-size IoT system.

## 5. Experiments and Tests

### 5.1. Test Network Structure

IoT-Crypto solution was tested in the conditions imitating real-world IoT deployments. Test network consists of:Raspberry Pi 4B device performing the role of an IoT gateway,two Raspberry Pi 4B devices working as IoT devices,Wi-Fi router providing the Internet access and connectivity between devices, depending on the tested configuration,cloud virtual machine performing the role of IoT-Crypto cloud server.

The test network corresponds to the IoT network structure described in Section 3.1. It allows to test all the features of the IoT-Crypto solution and to perform performance measurements. It is also possible to reconfigure the network and test various wireless communication standards.

The gateway and IoT devices constituting a single IoT subnet are typically located in close physical proximity and close network neighbourhood. The cloud server may be, however, distant. Using a cloud virtual machine allows reproducing the actual network conditions, especially connection stability and delays. Alternatively, it would be possible to emulate those conditions artificially.

### 5.2. Protocol Overhead Related to the Cryptographic Techniques

Securing the IoT network brings along several types of transmission and computation overhead. First of them is the initial handshake mechanism needed to establish a secure communication channel. It involves transmitting 1639 bytes of data in the IoT-Crypto network. This amount includes exchanging certificates and can be considered relatively small, given that a single X.509 certificate is often bigger than 2000 bytes.

The encryption of the data causes the second type of overhead. It was measured that a single encrypted UDP packet sent by IoT-Crypto application is 29 bytes bigger compared to the transmission without encryption, as seen in the chart in Figure 8. It complies with the specification of DTLS v1.2 [15]. This overhead could be further reduced to 11 bytes per packet when using DTLS v1.3. This protocol version was not tested in the IoT-Crypto network, as the underlying MbedTLS library does not yet support it.

The communication between the devices was monitored using Wireshark network protocol analyser. The sizes of protocol messages and other data structures using various encoding methods were ascertained by running modified IoT-Crypto software. The modification involved adding support for encoding formats other than CBOR.

Comparable solutions employing DTLS, such as PKI4IoT [11], should have the same encryption overhead of 29 bytes per UDP packet. IoT-Crypto does not, however, use CoAP or HTTP, so it has a smaller overall protocol overhead. CoAP protocol header size is between 6 and 28 bytes for requests and 3 bytes for responses, as measured in the PKI4IoT network. IoT-Crypto does not introduce that type of overhead at all. Only a limited amount of information about certificate and message sizes or communication overhead is available for the PKI4IoT system. According to the presented results, the uncompressed certificate size is approximately 400 bytes. IoT-Crypto certificate size is 232 bytes without compression. Certificates are transmitted during the initial handshake mechanism, so their size directly impacts protocol overhead.

Another type is the computation overhead. It involves the time and resources needed to perform cryptographic operations compared to the situation with disabled encryption. Raspberry Pi 4B devices used during tests are equipped with quad-core ARM CPU and 2GB of RAM. This specification can be considered more comparable to the PCs than more constrained IoT devices. Simple observation (with basic power meter device and Linux CPU usage monitoring) did not show CPU usage and power consumption to be noticeably different when running IoT-Crypto software with and without cryptography mechanisms. More detailed measurements were performed using a C profiler. Their results are presented in the chart in Figure 9. They have shown that peer certificate verification is responsible for about half of the computation overhead.

### 5.3. Comparison of Encoding

CBOR format is used for encoding data transmitted in the IoT-Crypto network and stored on devices. It is a binary format with a structure similar to the text-based JSON. CBOR additionally has more strict type control than JSON.

CBOR is a recommended encoding standard for CoAP protocol. CoAP is a more lightweight alternative for HTTP, which commonly uses JSON encoding. IoT-Crypto communication protocol is custom built and is not based on any application layer protocol, such as CoAP or HTTP. CBOR encoding was chosen as a solution designed for constrained systems. It has also proven to be an optimal choice when directly compared with JSON and raw binary format in this particular application (see the comparison in Figure 10).

It is worth mentioning that CBOR-encoded data stored in binary format has a similar size to the raw data structures of the C language. It is even more efficient in some cases, thanks to more compact data representation and the non-optimal layout of corresponding C structures.

### 5.4. Suitability of the Wireless Standards

Various wireless communication standards may be used in IoT networks. Testing and comparing them was not the objective of this work. IoT-Crypto network requires using standards that support TCP/IP protocol stack and allow transmitting IP packets. The test network described in Section 5.1 was initially run using Ethernet wired connections and Wi-Fi wireless standards. Both are well-known, and their features and performance have already been thoroughly tested and ascertained.

The Wi-Fi wireless standard used during the tests is not well-suited for constrained IoT networks, as it was designed for other purposes. Ascertaining the best IoT wireless standard should be a part of a separate study. However, it is reasonable to test at least one of those standards in the IoT-Crypto network and ascertain its performance and suitability. The test network described in Section 5.1 allowed to test Bluetooth Low Energy (BLE) Internet Protocol Support Profile (IPSP). It was chosen because BLE is aimed at constrained IoT devices, and its IPSP provides support for the TCP/IP protocol stack.

6LowPAN was initially developed to transmit IPv6 packets in the IEEE 802.15.4 networks [17]. This functionality is achieved through header compression and adaptation layer, which performs fragmentation of too big packets. IPSP profile is an adaptation of the 6LowPAN standard for use in BLE networks that replaces the GATT protocol stack and directly uses the BLE link control layer (L2CAP). 6LowPAN over BLE is a very interesting wireless standard for two reasons. Firstly, it uses IPv6, which allows each device to have a public IP address related to the network interface identifier. Secondly, the underlying BLE standard was explicitly designed for IoT purposes and is appropriate for constrained devices.

IoT-Crypto software was modified to be able to use IPv6 instead of IPv4. One of the Raspberry Pi devices in the test network was configured to act as an IPv6 router and gateway for 6LowPAN BLE devices. No issues arose when operating the IoT-Crypto network in that configuration. Additional measurements were performed to ascertain connection performance. Maximum connection throughput achieved during tests is 239.2 kbit/s at the 1 m distance between devices (see chart in Figure 11). Throughput is significantly smaller in close proximity (possibly because of interferences), but it is inversely proportional to more considerable distances. Expected BLE throughput depends on many factors, such as transmission interval, radio interference and application-layer protocol overhead. Measured values are within the expected range for BLE, as described in the BLE performance evaluation by Tosi et al. [18].

6LowPAN BLE connection on the Raspberry devices did not work correctly when more than one device was connected to the gateway device. Raspbian OS based on version 4.19 of the Linux kernel was installed on the devices. A bug was identified in the Linux kernel source code. The destination address was not resolved correctly in the situation mentioned above. Fix was implemented in the kernel method responsible for the peer address lookup. Next, all tests and measurements were performed using the Linux kernel with that fix.

Measured connection round-trip times are much bigger than those known from the Internet network and Wi-Fi connections. There was a vast difference between the minimum and maximum results. The chart in Figure 12 presents the average values of 200 results obtained in each configuration. Tests were performed without changing the default configuration of the devices. Default BLE connection interval is between 30 ms (minimum connection interval) and 60 ms (maximum connection interval). It means that data packets are not transmitted immediately when they are ready but only during data transfer events, which occur with a configured interval. Measured values conform with the expectations resulting from the connection settings. For one-to-one connection, the delay caused by the connection interval can be as big as 120 ms. For the connection with an intermediary device, it can reach 180 ms. Obtained results are generally consistent with those presented in the detailed study on IPv6-Based BLE connectivity by Darroudi et al. [19]. However, that study focused on Contiki OS implementation of the protocol stack and did not take into account differences caused by the distance between devices.

It has proven to be impossible to enable connection encryption. Data are transmitted as plain text. Linux kernel documentation does not mention this issue, and the source code does not suggest this functionality is implemented. A data encryption mechanism must be implemented in the application layer (like in the IoT-Crypto system).

Connection characteristics prove that 6LowPAN BLE may be used in the IoT-Crypto, and more generally, IoT network. Throughput is adequate for transmitting small amounts of data, and delays are sufficient for most applications, which do not require real-time processing. The results highlight the reasons for BLE power efficiency when compared to Wi-Fi. Data are transmitted periodically instead of continuously, which causes more significant delays. Additionally, less power is required for smaller throughputs.

## 6. Conclusions and Future Work

In this paper, we presented IoT-Crypto—a custom-designed IoT solution that can be used to build a fully functional and secure IoT network. IoT-Crypto design was based on the identified traits and structure of the IoT systems, taking into account the most common architectural aspects, data flows, hardware constraints, and security requirements.

The presented solution was built from scratch. This approach allowed far-reaching optimisation, thus avoiding potential problems caused by using common Internet standards that have not been designed for IoT purposes. IoT-Crypto contains more lightweight alternatives for X.509 certificates and uses custom application layer protocol instead of multi-purpose HTTP and CoAP. It also uses pre-existing and proven protocols, such as DTLS, providing the basis for cryptography and connection security and CBOR—a very efficient data serialisation standard.

The most original and innovative part of the IoT-Crypto solution is the cryptographic relation of trust, modelled on the observed business relations between entities operating IoT networks. It allows high flexibility and good configuration capabilities. This trust model was achieved thanks to the custom certificate format, which is loosely based on X.509 and OpenPGP alternatives.

The other very distinctive trait of the IoT-Crypto is a close link between autoconfiguration mechanisms and cryptography. Various aspects of network functioning, such as device detection, removing and ascertaining network structure, directly use cryptography. IoT-Crypto is security-driven. It was built around cryptography techniques rather than trying to make pre-existing communication mechanisms secure.

IoT-Crypto software can be run on a variety of hardware platforms. It is written in C (except for part of cloud server software), which allows running it on almost all devices equipped with the TCP/IP stack support. Tests and experiments verified the proper functioning of the designed mechanism in the test network.

Future work shall concentrate on in-depth security analysis of the IoT-Crypto network, along with penetration tests and vulnerability scans. Moreover, for improving efficiency, it is also possible to use the hardware acceleration of cryptographic operations. Some of the commercially available microcontrollers may provide such functionality. Technical report on the Texas Instruments SimpleLink CC13x2/CC26x2 models [20] shows that SHA-256 calculation for 8 kB of data can be forty times faster than the software implementation. IoT-Crypto uses the SHA-256 algorithm during session key calculation so that it would be a noticeable improvement. Elliptic key Diffie-Hellmann shared key calculation using hardware acceleration is eleven times faster. Both results were obtained in direct comparison with the mbedTLS library, which is the core of IoT-Crypto software. Therefore, it is proven that such an improvement is feasible to achieve in the IoT-Crypto network.

A further energy saving can be achieved by UDP/IP, and DTLS header compression for packets exchanged between the IoT device and the gateway, e.g., as the eeDTLS protocol proposed by Banerjee et al. [21]. eeDTLS reduces protocol overheads by 91%, from 77 bytes to 7 bytes. eeDTLS also allows a client to send certificate URLs in place of certificates according to RFC 6066. In consequence, eeDTLS can reduce energy consumption by DTLS handshake computations.

Another aspect is creating a user-friendly GUI. Currently, software running on the cloud server exposes REST API. It was not in-depth discussed in this work, as it is not directly related to the security mechanisms of the network. It allows interacting with the network—fetching information about network structure, issuing commands, fetching sensor data. This API can be used as a base for creating GUI that will allow convenient network monitoring and management. The third direction of future work includes performing a comprehensive comparative analysis of the IoT-Crypto and other IoT solutions and evaluating standards and protocols that may be potentially used in the IoT-Crypto network. All this work will help improve IoT-Crypto, identify and fix potential problems. It will also contribute to IoT research, helping identify the best solutions and practices.

## Figures and Tables

**Figure 1 sensors-21-06906-f001:**
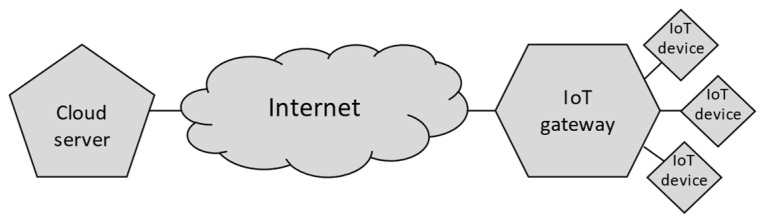
The general architecture of IoT-Crypto network.

**Figure 2 sensors-21-06906-f002:**
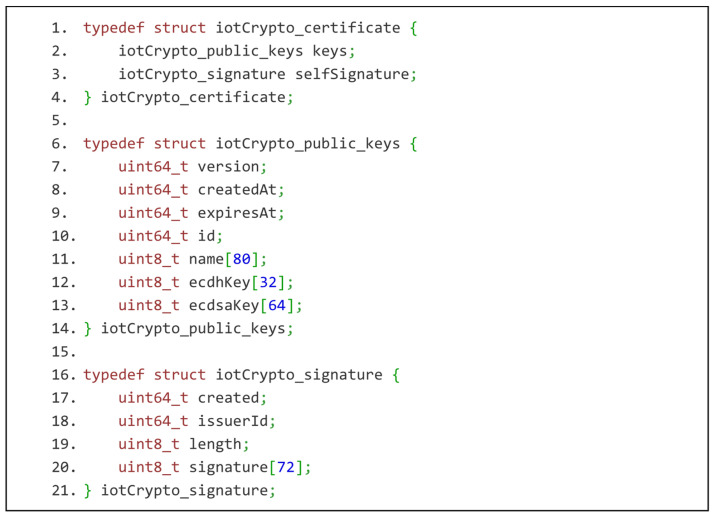
IoT-Crypto certificate as C structures. The certificate consists of public keys with their metadata and self-signature. Third-party signatures have the same structure as self-signature. They are, however, independent entities stored and transmitted separately from the certificate.

**Figure 3 sensors-21-06906-f003:**
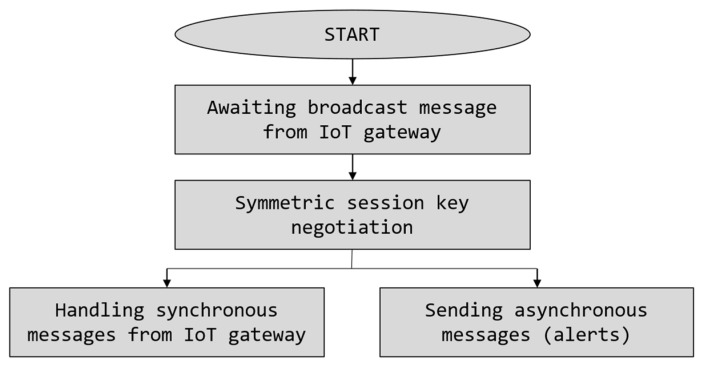
IoT device software operation in the IoT-Crypto system.

**Figure 4 sensors-21-06906-f004:**
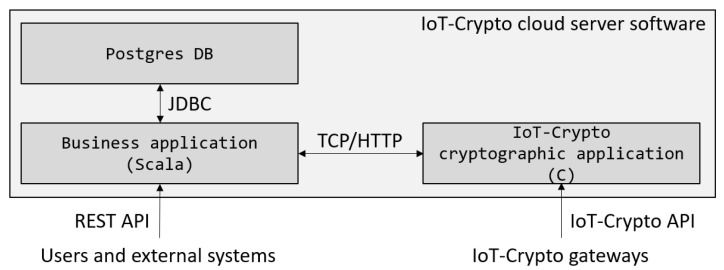
IoT-crypto server software structure.

**Figure 5 sensors-21-06906-f005:**
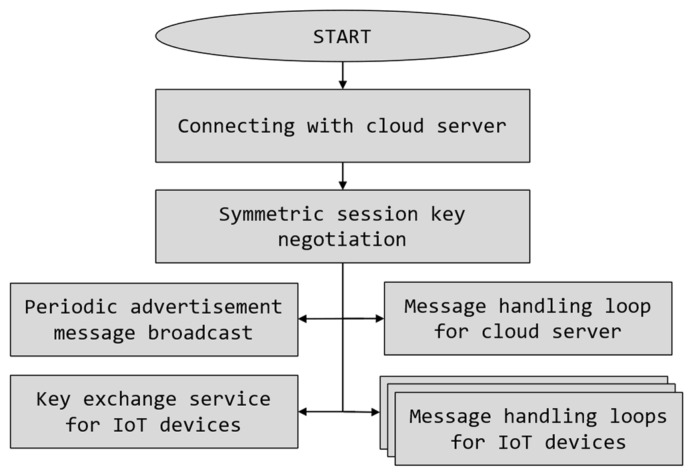
IoT gateway software operation in the IoT-Crypto system.

**Figure 6 sensors-21-06906-f006:**
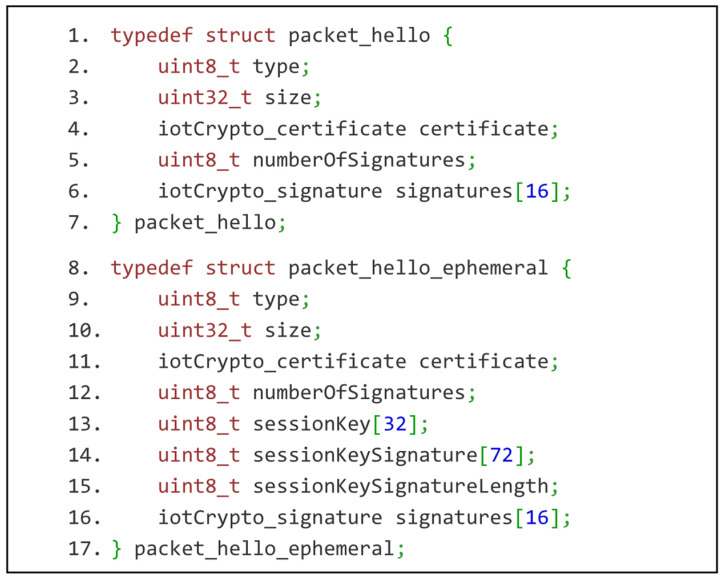
IoT-Connection-Protocol messages presented as C structures.

**Figure 7 sensors-21-06906-f007:**
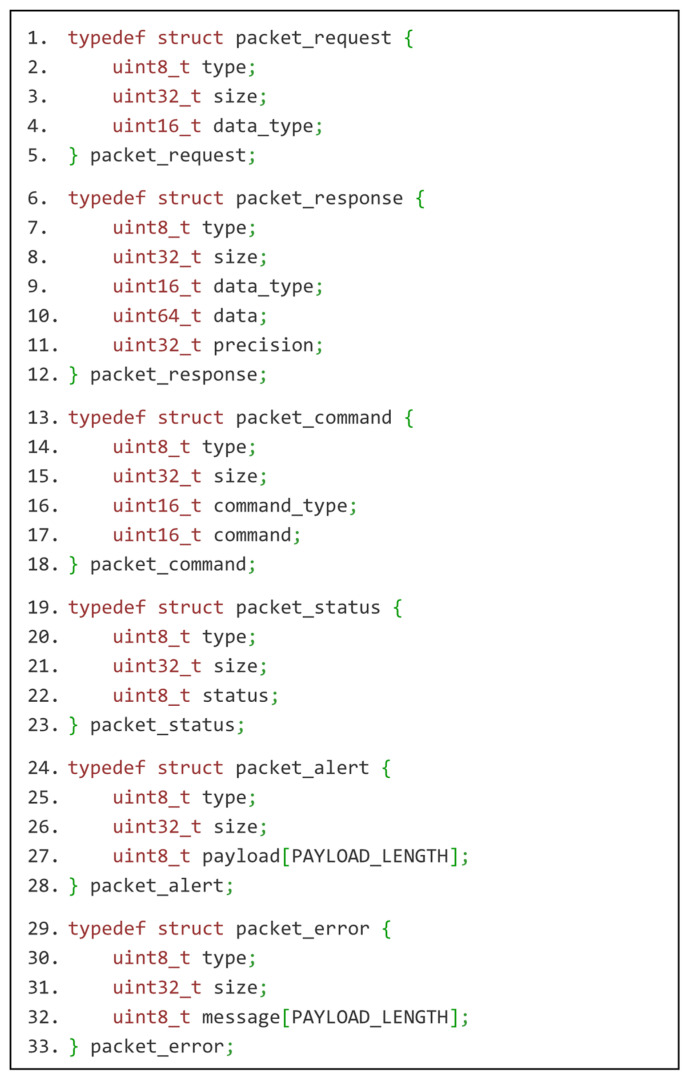
IoT-Transmission-Protocol messages presented as C structures (sensor reading request, sensor reading response, actuator command, command execution status, alert message and error message).

**Figure 8 sensors-21-06906-f008:**
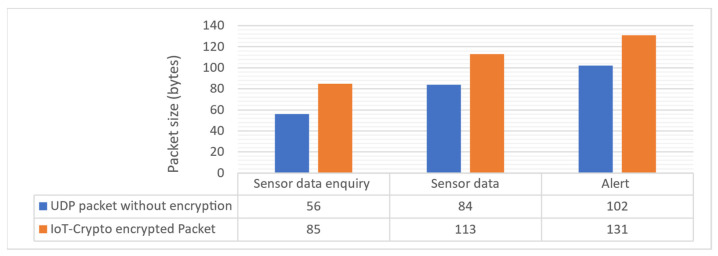
Sizes of IoT-Crypto transmission protocol packets.

**Figure 9 sensors-21-06906-f009:**
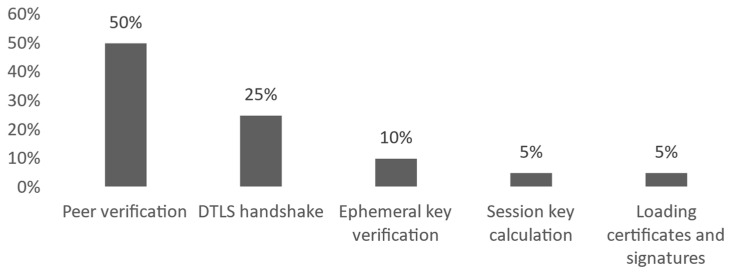
IoT-Crypto computation overhead divided into categories.

**Figure 10 sensors-21-06906-f010:**
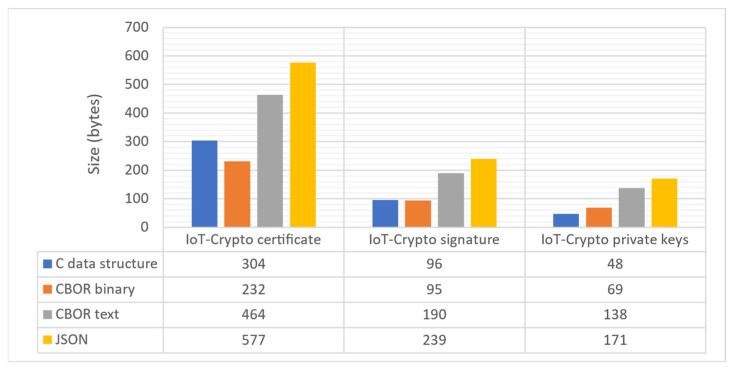
Sizes of IoT-Crypto data structures using various encodings (CBOR data can be stored in binary format or as a hexadecimal string, which is twice as long).

**Figure 11 sensors-21-06906-f011:**
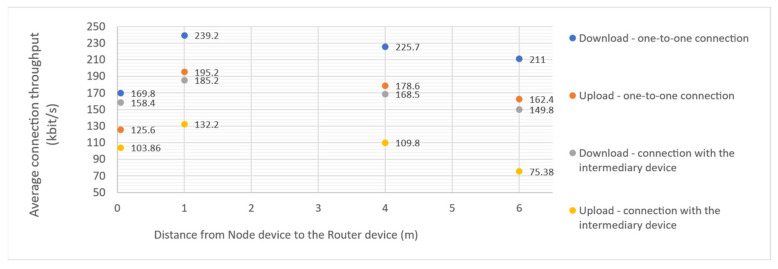
6LowPAN BLE average connection throughput measured in the test network using iperf3 software. The measurement was performed 50 times in each configuration. The obtained results were similar and repetitive. The points presented on the chart correspond to the averages of those results.

**Figure 12 sensors-21-06906-f012:**
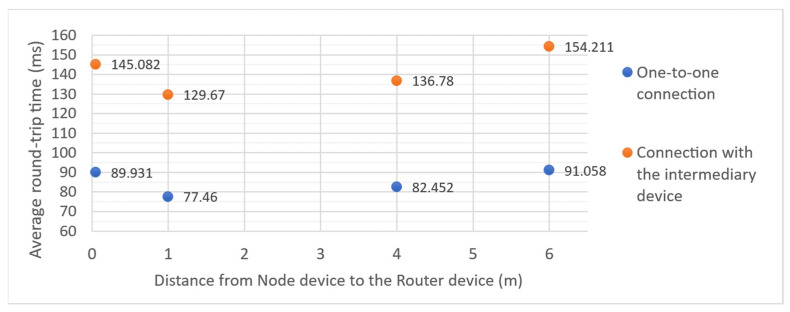
6LowPAN BLE transmission delays measured in the test network using Ping network utility. The measurement was performed 200 times in each configuration. The points presented on the chart correspond to the averages of the obtained results.

## Data Availability

Not applicable.

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
