# Peer review of "A Secure Communication System for Constrained IoT Devices—Experiences and Recommendations"

_sensors, 2021, doi:10.3390/s21206906_

Round 1

Reviewer 1 Report

The authors present a system for secure communications in an IoT environment, making a good state of the art study, describing the implemented system in detail and giving examples of C codes used. On the other hand, they give a detailed schematic explanation of the software and clearly present the results of the tests carried out.

It would be interesting to add more references so that readers can research and have more overview of the work done.

Author Response

Response to Reviewer 1 Comments

Point 1: It would be interesting to add more references so that readers can research and have more overview of the work done. 

Response 1: Thank you for the review.

We described 12 related works (almost two entire pages). Our references point to documents closely related to our work. Our paper does not analyse existing publications about secure communication in IoT systems, which is a vast domain. Thus, in our opinion, the number of references is adequate to the paper content. Anyway, we added one reference pointing a possible direction of further IoT-Crypto improvement. The added paragraph on page 17 we cite below.

A further energy saving can be achieved by UDP/IP, and DTLS header compression for packets exchanged between the IoT device and the gateway, e.g. as the eeDTLS protocol proposed by Banerjee et al. [17]. eeDTLS reduces protocol overheads by 91%, from 77 bytes to 7 bytes. eeDTLS also allows a client to send certificate URLs in place of certificates according to RFC 6066. In consequence, eeDTLS can reduce energy consumption by DTLS handshake computations.

Sincerely yours

Michał Goworko and Jacek Wytrębowicz

Reviewer 2 Report

The authors have presented a lightweight certificate format for IoT along with an IoT system implementation. The paper is overall well-written, and this is an important research direction in the context of efficient and secure IoT.

Few comments:

- the IoT security protocol implementation in this work is based on transport layer security (TLS), please mention this in the abstract

- the TLS software implementation is based on open-source mbedTLS library, please add corresponding reference

Ref: https://github.com/ARMmbed/mbedtls

- please mention which TLS cipher suite has been used; the certificate contains ECDH public key and ECDSA public key (along with signature); so, is this a non-ephemeral key exchange with static Diffie-Hellman keys? also, what is the encryption algorithm and hash function used?

- the certificate is self-signed, with public key contained in the certificate itself ... self-signed certificates are not considered secure, what is the motivation behind this choice? ... please also include and analyze the scenario where the certificate is signed by a trusted certificate authority ... TLS allows caching of certificate authority public keys, so they need not be explicitly included in the certificate

- one of the objectives of this work is to reduce communication overhead in IoT security protocol; while lightweight certificates are considered here, there are several previous works which also consider another direction regarding packet compression; few additional references are provided below, recommend including them in the related work section for completeness:

Ref1: "6LoWPAN compressed DTLS for CoAP" - https://ieeexplore.ieee.org/abstract/document/6227754/

Ref2: "Lithe: Lightweight secure CoAP for the internet of things" - https://ieeexplore.ieee.org/abstract/document/6576185/

Ref3: "E-Lithe: A lightweight secure DTLS for IoT" - https://ieeexplore.ieee.org/abstract/document/8288362/

Ref4: "eeDTLS: Energy-efficient datagram transport layer security for the Internet of Things" https://ieeexplore.ieee.org/abstract/document/8255053/

Author Response

Response to Reviewer 2 Comments

Thank you for the review and constructive comments.

Point 1: the IoT security protocol implementation in this work is based on transport layer security (TLS), please mention this in the abstract.

Response 1: We added information about the IoT-Crypto protocol being based on DTLS in the abstract (as given below).

(…) IoT-Crypto introduces an innovative, lightweight certificate format and trust model based on real-world business relations. It also specifies secure communication protocol, which uses underlying encrypted DTLS connection. (…)

It was also already mentioned in Chapter 4.6 (page 11). We quote below a relevant section of the paragraph.

Communication in the IoT-Crypto network is based on the TCP/IP protocol stack with the UDP in the transport layer and MbedTLS implementation of DTLS protocol on the border between transport and application layers. The application layer consists of two custom protocols: IoT-Connection-Protocol and IoT-Transmission-Protocol. Comparable solutions, such as XIOT, often use CoAP or HTTP in that layer.

Point 2: the TLS software implementation is based on open-source mbedTLS library, please add corresponding reference

Ref: https://github.com/ARMmbed/mbedtls

Response 2: We added the reference in Chapter 4.2 (page 8).

Point 3: please mention which TLS cypher suite has been used; the certificate contains ECDH public key and ECDSA public key (along with signature); so, is this a non-ephemeral key exchange with static Diffie-Hellman keys? also, what is the encryption algorithm and hash function used?

Response 3: We added the information about the employed DTLS cypher suite, including encryption algorithm and hash function, in Chapter 4.6 on page 10 (the paragraph given below).

The underlying UDP connection is secured using DTLS and uses cypher suite TLS_PSK_WITH_CHACHA20_POLY1305_SHA256 (defined in the RFC 8439 [16]). It is a suite of cryptographic algorithms which consist of SHA256 hash function, Poly1305 message authentication code and ChaCha20 symmetric stream cypher. According to the specification, it can be up to three times faster than cypher suites that use AES cypher. Cryptographic key exchange is performed on the IoT-Crypto level, and DTLS uses the obtained key as a pre-shared key (PSK).

The key exchange is, in fact, ephemeral. One side of the transmission (less constrained of the two devices) must always generate and use ephemeral key pairs to achieve forward security. We think that this mechanism is described sufficiently in Chapter 4.6 (on page 11, line 459). We quote below a relevant section of the already existing paragraph.

(…) The session key derived from the elliptic Diffie–Hellman key exchange would be the same for each connection between the same devices. Forward security is achieved by making one side of the transmission use ephemeral keys. Less constrained of the two connecting devices is responsible for generating the ephemeral key pair (gateway when connecting with IoT devices and cloud server when connecting with gateways).

Point 4: the certificate is self-signed, with public key contained in the certificate itself ... self-signed certificates are not considered secure, what is the motivation behind this choice?

Response 4: This aspect of the IoT-Crypto was inspired by OpenPGP PKI. IoT-Crypto certificate may have many signatures issued by third parties, but on the other hand, it does not have to be signed by any third party. It is self-signed to ensure its integrity and prove the possession of the corresponding private key (the same as OpenPGP certificates).

Indeed, self-signed certificates are not considered secure. But that problem does not concern IoT-Crypto. The IoT-Crypto certificate will be verified as trusted only when signed by a trusted third party (when one of its many signatures was issued by a party that the verifier trusts).

We edited the relevant paragraph in Chapter 3.2 (page 5) to be more clear and better describe this mechanism. We quote below the modified paragraph.

Each certificate may possess signatures generated by other parties. Third-party signatures are associated with the certificate but are not strictly part of it. Therefore, each certificate is self-signed to ensure its integrity and prove possession of the corresponding private key. Additional signatures may be stored alongside the certificate on the device they pertain to or may be known only to the verifying party and stored separately. The only condition is that they must be available during certificate verification. This mechanism shows similarities with the OpenPGP PKI.

Additionally, we modified the caption of Figure 2. (page 7) to explain that the third-party signatures have the same structure as self-signature but are stored and transmitted separately from the certificate. We quote the modified caption below.

Figure 2. IoT-Crypto certificate as C structures. The certificate consists of public keys with their metadata and self-signature. Third-party signatures have the same structure as self-signature. They are, however, independent entities that are stored and transmitted separately from the certificate.

Point 5: please also include and analyze the scenario where the certificate is signed by a trusted certificate authority ... TLS allows caching of certificate authority public keys, so they need not be explicitly included in the certificate.

Response 5: Certificate authority public keys are not included in the IoT-Crypto certificate, as is shown in Figure 2. They are also not transmitted during the key exchange, as shown in Figure 6.

All described situations and scenarios already apply to the case when the certificate is signed by a trusted third party. The IoT-Crypto certificate will be verified as trusted only when a trusted third party signs it. If the certificate is only self-signed and does not have any signatures issued by a trusted third party, it will be rejected, and it would not be possible to establish the connection.

We think that the relationship of trust in the IoT-Crypto network is sufficiently described in Chapter 3.2 (page 5, line 266). We quote below the relevant section.

The trust model of the IoT-Crypto system is inspired by both X.509 and OpenPGP. Relation of trust is not transitive, as in X.509 model. It is decentralized to correspond with business relations between companies – owners and operators of IoT networks. Many other parties may then sign each certificate. A certificate is trusted when at least one of the signatures was issued by a trusted party. IoT-Crypto trust model allows dynamic changes of the scope of cooperation between companies by modifying the list of trusted certificates and the list of signatures associated with certificates.

Point 6: one of the objectives of this work is to reduce communication overhead in IoT security protocol; while lightweight certificates are considered here, there are several previous works which also consider another direction regarding packet compression; few additional references are provided below, recommend including them in the related work section for completeness:

Ref1: "6LoWPAN compressed DTLS for CoAP" - https://ieeexplore.ieee.org/abstract/document/6227754/

Ref2: "Lithe: Lightweight secure CoAP for the internet of things" - https://ieeexplore.ieee.org/abstract/document/6576185/

Ref3: "E-Lithe: A lightweight secure DTLS for IoT" - https://ieeexplore.ieee.org/abstract/document/8288362/

Ref4: "eeDTLS: Energy-efficient datagram transport layer security for the Internet of Things" https://ieeexplore.ieee.org/abstract/document/8255053/

Response 6: Thank you for indicating the interesting and valuable articles.

We described 12 related works (almost two entire pages). Our references point to documents closely related to our work. Our paper does not analyse existing publications about secure communication in IoT systems, which is a vast domain. Thus, in our opinion, the number of references is adequate to the paper content. Anyway, we added one reference pointing a possible direction of further IoT-Crypto improvement. The added paragraph on page 17 we quote below.

A further energy saving can be achieved by UDP/IP, and DTLS header compression for packets exchanged between the IoT device and the gateway, e.g. as the eeDTLS protocol proposed by Banerjee et al. [17]. eeDTLS reduces protocol overheads by 91%, from 77 bytes to 7 bytes. eeDTLS also allows a client to send certificate URLs in place of cer-tificates according to RFC 6066. In consequence, eeDTLS can reduce energy consumption by DTLS handshake computations.

Sincerely yours

Michał Goworko and Jacek Wytrębowicz

Reviewer 3 Report

Contributions:

The IoT concept has been swiftly developing and gaining popularity in recent years. IoT systems collect and process vast amounts of often sensitive data. Information security should be the key feature of an IoT network. In this paper, IoT-Cryptosecure communication system for the Internet of Things is presented. It addresses IoT features, such as constrained abilities of devices, needs to reduce the volume of the transmitted data and be compatible with the Internet.

This paper conducts tests in the conditions imitating real-world IoT deployments. Results of tests and experiments performed in the IoT-Crypto network confirm that it works correctly and securely. Test network was also used to ascertain the suitability of encoding standards and BLE IPSP profile for the IoT.

Strengths:

The presented solution has been built from scratch. This approach allowed far-reaching optimization, thus avoiding potential problems caused by using common Internet standards that have not been designed for IoT purposes

Detailed Comments:

1) In 4.3 ”IoT-Crypto device software”, Software of the IoT device in the IoT-Crypto system can process synchronous messages and asynchronous messages simultaneously. This may require a more detailed explanation.

2) CBOR format is used for encoding data transmitted in the IoT-Crypto network and stored on devices. Sizes of IoT-Crypto data structures using various encodings are different in Figure 10. Is there a need to further check the frequency of use of certificates, signatures and private keys?

3) The article concludes by examining the suitability of wireless communication using the 6LowPAN specification. Is the main scope of this work 6LowPAN BLE connection? Is this work applicable to other wireless communication standards?

4)Securing and privacy for IOT has already widely investigated. The authors fail to properly cite and carefully discuss the differences between this paper and several prior work highly related to it (e.g., [1-3]).

[1] Securely Connecting Wearables to Ambient Displays with User Intent, TDSC 2020.

[2] Incentive Mechanism for Privacy-Aware Data Aggregation in Mobile Crowd Sensing Systems, TON 2018.

[3] Injecting Reliable Radio Frequency Fingerprints Using Metasurface for the Internet of Things, TIFS 2021

5) The paper is mainly about a secure communication system. But the experiment does not test its security. Its security needs more experiments.

Author Response

Response to Reviewer 3 Comments

Thank you for the review and constructive comments.

Point 1: In 4.3 ”IoT-Crypto device software”, Software of the IoT device in the IoT-Crypto system can process synchronous messages and asynchronous messages simultaneously. This may require a more detailed explanation.

Response 1: We modified the relevant paragraph to be more clear and precise (page 8, line 375). We quote it below.

(…) After successful key negotiation, the device enters event handling mode and processes synchronous (request/response) messages received from the gateway, i.e. sensor reading requests and commands. The IoT device may also send asynchronous (spontaneous) messages to the gateway, e.g. alerts based on the sensor readings. These two tasks are running simultaneously in separate threads.

Point 2: CBOR format is used for encoding data transmitted in the IoT-Crypto network and stored on devices. Sizes of IoT-Crypto data structures using various encodings are different in Figure 10. Is there a need to further check the frequency of use of certificates, signatures and private keys?

Response 2: We compared various encoding methods, but IoT-Crypto only uses binary CBOR encoding, as described in Chapter 4.6. “IoT-Crypto communication protocol”.

Certificates, signatures, and private keys are only used in a way generally accepted among the solutions using asymmetric cryptography. It means that certificates and signatures are only transmitted during key exchange. Private keys are never transmitted and are only stored on corresponding devices. It follows that the frequency of use of certificates is comparable to other similar solutions. We believe it does not require further analysis.

Point 3: The article concludes by examining the suitability of wireless communication using the 6LowPAN specification. Is the main scope of this work 6LowPAN BLE connection? Is this work applicable to other wireless communication standards?

Response 3: Any wired and wireless standard supporting TCP/IP network protocol stack may be used in the IoT-Crypto network. 6LowPAN BLE was tested as a real-world example of wireless protocol aimed at IoT networks. The relevant paragraphs in Chapter 5.4 (page 15) were modified to be more explicit and better explain this motivation. We quote them below.

Various wireless communication standards may be used in IoT networks. Testing and comparing them was not the objective of this work. IoT-Crypto network requires using standards that support TCP/IP protocol stack and allow transmitting IP packets. The test network described in Chapter 5.1 was initially run using Ethernet wired connections and Wi-Fi wireless standards. Both are well-known, and their features and performance have already been thoroughly tested and ascertained.

The Wi-Fi wireless standard used during the tests is not well-suited for IoT networks, as it has been designed for other purposes. Ascertaining the best IoT wireless standard should be a part of a separate study. The characteristics of wireless standards aimed at IoT and Wi-Fi differ substantially. Therefore, it is reasonable to test at least one of those standards in the IoT-Crypto network and ascertain its performance and suitability. The test network described in Chapter 5.1 allowed to test Bluetooth Low Energy (BLE) Internet Protocol Support Profile (IPSP). It was chosen because BLE is aimed at constrained IoT devices, and its IPSP provides support for the TCP/IP protocol stack.

Point 4: Securing and privacy for IOT has already widely investigated. The authors fail to properly cite and carefully discuss the differences between this paper and several prior work highly related to it (e.g., [1-3]).

[1] Securely Connecting Wearables to Ambient Displays with User Intent, TDSC 2020.

[2] Incentive Mechanism for Privacy-Aware Data Aggregation in Mobile Crowd Sensing Systems, TON 2018.

[3] Injecting Reliable Radio Frequency Fingerprints Using Metasurface for the Internet of Things, TIFS 2021

Response 4: Thank you for indicating the interesting and valuable articles.

The security of communication in IoT systems is a vast domain. It comprises many issues, bootstrapping, firmware/software updates, trust for data and services, data privacy, radio link security, to mention a few of them. We do not find the indicated articles directly related to our work.

We described 12 related works (almost two entire pages). Our references point to documents closely related to our work, which we consider valuable, and we can recommend for the reader. Our paper does not analyse existing publications about secure communication in IoT systems. Thus, in our opinion, the number of references is adequate to the paper content. Anyway, we added one reference pointing a possible direction of further IoT-Crypto improvement. The added paragraph on page 17 we cite below.

A further energy saving can be achieved by UDP/IP, and DTLS header compression for packets exchanged between the IoT device and the gateway, e.g. as the eeDTLS protocol proposed by Banerjee et al. [17]. eeDTLS reduces protocol overheads by 91%, from 77 bytes to 7 bytes. eeDTLS also allows a client to send certificate URLs in place of cer-tificates according to RFC 6066. In consequence, eeDTLS can reduce energy consumption by DTLS handshake computations.

Point 5: The paper is mainly about a secure communication system. But the experiment does not test its security. Its security needs more experiments.

Response 5: We have aimed to present key aspects of the design and experiences from implementing the IoT-Crypto, and to analyse its performance. The experiments on its resistance against different attack vectors could be done as the next step. However, we can expect that the results from such experiments will confirm the known security strength of the TLS/DTLS mechanisms based on digital certificates.

Sincerely yours

Michał Goworko and Jacek Wytrębowicz

Reviewer 4 Report

 In this paper, the authors present the IoT-Crypto – secure communication system for the Internet of Things. 
The study is interesting and motivations are good. However, the paper is not well structured; the methodology section is too short and not clear. 
The literature review section presents useful cited papers.

I am confident if the authors resubmit the paper by reviewing the methodology, the document would acquire more quality.

Author Response

Response to Reviewer 4 Comments

Point 1: the paper is not well structured; the methodology section is too short and not clear… I am confident if the authors resubmit the paper by reviewing the methodology, the document would acquire more quality.

Response 1: Thank you for the review.

We added an explanation in the section that pertains to the protocol overhead analysis (page 14, line 539). We quote it below.

The communication between the devices has been monitored using a Wireshark network protocol analyser. The sizes of protocol messages and other data structures using various encoding methods have been ascertained by running modified IoT-Crypto software. The modification involved adding support for encoding formats other than CBOR. 

We modified the article section describing the 6LowPAN BLE tests (page 15, line 589) to be more explicit and better explain the motivation behind performing those tests. We quote the revised fragment below.

Various wireless communication standards may be used in IoT networks. Testing and comparing them was not the objective of this work. IoT-Crypto network requires using standards that support TCP/IP protocol stack and allow transmitting IP packets. The test network described in Chapter 5.1 was initially run using Ethernet wired connections and Wi-Fi wireless standard. Both are well-known, and their features and performance have already been thoroughly tested and ascertained. 

The Wi-Fi wireless standard used during the tests is not well-suited for IoT networks, as it has been designed for other purposes. Ascertaining the best IoT wireless standard should be a part of a separate study. The characteristics of wireless standards aimed at IoT and Wi-Fi differ substantially. Therefore, it is reasonable to test at least one of those standards in the IoT-Crypto network and ascertain its performance and suitability. The test network described in Chapter 5.1 allowed to test Bluetooth Low Energy (BLE) Internet Protocol Support Profile (IPSP). It was chosen because BLE is aimed at constrained IoT devices, and its IPSP provides support for the TCP/IP protocol stack. 

We also added additional explanations of the methodology of the performance measurements in the captions of Figures 11 and 12.

Sincerely yours

Michał Goworko and Jacek Wytrębowicz

Round 2

Reviewer 3 Report

comments addressed 

Reviewer 4 Report

The paper now is ready to be published in the journal.